# Psychotherapeutic interventions for burns patients and the potential use with Stevens-Johnson syndrome and toxic epidermal necrolysis patients: A systematic integrative review

Pauline O'Reilly[1,2,3]*, Pauline Meskell[1,2], Barbara Whelan[1,2], Catriona Kennedy[1,4], Bart Ramsay[5,6], Alice Coffey[1,2,3], Donal G. Fortune[2,7], Sarah Walsh[8], Saskia Ingen-Housz-Oro[9,10,11], Christopher B. Bunker[12], Donna M. Wilson[1,13], Isabelle Delaunois[14], Liz Dore[15], Siobhan Howard[2,7], Sheila Ryan[1,5]

1 Department of Nursing and Midwifery, University of Limerick, Limerick, Ireland, 2 Health Research Institute, University of Limerick, Limerick, Ireland, 3 Health Implementation Science and Technology (HIST) Research Cluster, University of Limerick, Limerick, Ireland, 4 School of Nursing, Midwifery and Paramedic Practice, Robert Gordon University, Aberdeen, Scotland, 5 Charles Centre for Dermatology, University Hospital Limerick, ULHG, Limerick, Ireland, 6 School of Medicine, University of Limerick, Limerick, Ireland, 7 Department of Psychology, University of Limerick, Limerick, Ireland, 8 Dermatology Department, King's College Hospital, London, United Kingdom, 9 Dermatology Department, AP-HP, Henri Mondor Hospital, Créteil, France, 10 Reference Center for Toxic Bullous Dermatoses and Severe Drug Reactions TOXIBUL, Créteil, France, 11 Univ Paris Est Créteil EpiDermE, Créteil, France, 12 Department of Dermatology, University College London Hospitals NHS Foundation Trust, London, United Kingdom, 13 Faculty of Nursing, University of Alberta, Edmonton, Alberta, Canada, 14 Regional Medical Library, University Hospital Limerick, Limerick, Ireland, 15 Glucksman Library, University of Limerick, Limerick, Ireland

☯ These authors contributed equally to this work.
* Pauline.OReilly@ul.ie

## Abstract

### Background

The existing evidence demonstrates that survivors of SJS/TEN have reported long-lasting psychological effects of their condition. Burns patients experience similar psychological effects. It is important to look at ways to help allay the psychological complications of SJS/TEN. As there is an absence of evidence on SJS/TEN psychotherapeutic interventions, it was judged to be beneficial to determine the evidence underpinning psychotherapeutic interventions used with burns patients.

### Aims and objectives

The aim of this systematic integrative review was to synthesize the evidence relating to psychotherapeutic interventions used with adult burns patients and patients with SJS/TEN.

### Method

The systematic review was guided by Whittemore and Knafl's integrative review process and the PRISMA guidelines. Nine databases were searched for English and French

**Data Availability Statement:** All relevant data are within the article and its Supporting Information files.

**Funding:** This research was funded by the Health Research Institute, University of Limerick, Limerick, Ireland. The funders had no role in study design, data collection and analysis, decision to publish, or preparation of the manuscript.

**Competing interests:** The authors have declared that no competing interests exist.

language papers published January 2008 to January 2021. The protocol for the review was registered with PROSPERO.

## Results

Following a screening process, 17 studies were included in the review. Two themes were identified using content analysis, (i) Empirically supported psychotherapeutic treatments, (ii) Alternative psychotherapeutic treatments. This review revealed no evidence on specific psychotherapeutic interventions for patients with SJS/TEN. Some of the interventions used with burns patients, viz. relaxation therapy, hypnosis and cognitive behavioral therapy showed some significant benefits. However, the evidence for burns patients is mainly focused on pain and pain anxiety as outcomes.

## Conclusion

Following further research, some of the interventions deployed in burns patients may be applicable to SJS/TEN patients, particularly stress reduction techniques. In addition, the caring behaviours such as compassion, respect, and getting to know the patient as a person are important components to psychological care.

## Introduction

The epidermal loss observed in Stevens-Johnson syndrome (SJS) and toxic epidermal necrolysis (TEN) is similar to that seen in patients with extensive second-degree burns Bastuji-Garin et al. [1]. Patients with SJS and TEN are frequently managed in burns units [2] SJS and TEN are severe mucocutaneous conditions usually occurring from a reaction to a medication [3]. Both conditions are on the same disease continuum, with the latter being a more extensive and severe version of the former [4]. In the early stages of the disease trajectory, the patient may become critically ill, present with high fever, cutaneous erythema, and develop blisters [5]. Due to the severity of the condition, the focus of care is on ensuring patient survival. Health care practitioners have referred to the importance of advocating for their patient to ensure that they are admitted to an Intensive Care Unit/Burns Unit [6, 7]. Although the incidence rate of SJS/TEN is frequently understood as rare, 1–2 cases per million per year globally [8, 9], recent evidence highlights that incidence rates may be as high as 6.5 cases per million population per year [10]. Moreover, Chaby et al. [10] estimate that the annual mortality rate is 0.9 per million population. The mortality outcome for SJS/TEN patients has been shown to improve if they are cared for within an Intensive Care or Burns Unit [3, 11].

For survivors of SJS/TEN, there are many devastating outcomes. Some of the most common are cutaneous and ocular sequalae [12]. Respiratory and gastrointestinal tracts may also be affected [13]. There may also be long lasting psychological effects on patients and their significant others [14]. Whilst there is some awareness that psychological sequelae occur in patients there is a need to prioritise research in this area [13]. The existing evidence outlines that survivors of SJS/TEN report fear over taking medications [15–17], have self-image difficulties [16]; and quality of life concerns [18, 19]. In an observational study, Ingen-Housz-Oro et al. [20] found that SJS/TEN had a long-term impact on survivors and highlighted the importance of follow-up care post the acute phase.

Furthermore, some SJS/TEN patients sustain feelings of distress and reveal Post-Traumatic Stress (PTS) symptoms [18, 21], while others experienced anxiety and depression [17–19]. A recent study of SJS/TEN survivors (n = 121) found that 43.3% of participants screened positive for anxiety, 53.3% presented with a positive depression screen, and 19.5% had positive screening for PTS [22]. Dodiuk-Gad et al. [18] in their study of SJS/TEN survivors (n = 17), found that even though many of the individuals presented with psychological sequelae only four were assessed by a mental health practitioner following diagnosis with SJS/TEN. Similarly, in their review, Shanbhag et al. [23] outline that the psychological impact of SJS/TEN is insufficiently addressed and recommend that a psychiatric consultation should be offered to all patients. The French diagnosis and care protocol for SJS/TEN refer to the importance of providing psychological care which aims to reduce patient stress, including active prevention of Post-Traumatic Stress Disorder (PTSD) [11]. It is therefore important to look at ways to allay psychological complications occurring in these patients. Lee and Creamer [24] pose the question of whether psychological interventions in the acute phase of care may mitigate the occurrence of psychological problems among survivors of SJS/TEN.

As there is no evidence on the psychotherapeutic interventions used in the care of patients with SJS/TEN, it was judged to be beneficial to analyse and learn from the interventions used with burns patients. From a psychological perspective, both cohorts of patients may present with distress, anxiety and symptoms aligned with PTS [25, 26]. From a large prospective study of burn injury survivors (n = 1232) Fauerbach et al. [27] reported that significant psychological distress was present in 34% of the patients soon after hospital discharge. According to Wiechman and Patterson [28], during this critical phase the focus of psychological care is on helping the patient to cope with their situation with whatever defences or methods of coping they have. Non-pharmacological psychological interventions, such as hypnosis and relaxation, may help with pain control [28]. It is important to note that although SS/TEN and burns patients may present with similar psychological consequences, the specific supportive care guidelines for SJS/TEN differ from those used with burns [7]. For example, whilst the fluid, electrolyte, and nutrition management of patients with SJS/TEN is similar to the needs of burn patients [29], the fluid requirements are approximately 30% less in SJS/TEN patients compared to burn patients with similar levels of cutaneous involvement [7].

However, more evidence is required to understand how the best psychological care and support may be delivered to hospitalized patients with SJS/TEN. To address this issue, the following integrative review appraised the evidence relating to psychotherapeutic interventions that have been used with SJS/TEN (no evidence found) and burns patients during the acute stage of the illness, to reduce patients' feelings of stress and anxiety.

## Methods

### Aim

The aim of the integrative review was to explore and synthesize the evidence relating to psychotherapeutic interventions used with both adult patients with burns and those with SJS/TEN, during the acute stage of the illness. Examples of interventions that were considered for the review included cognitive behavioural therapy, hypnosis, meditation, psychotherapy, and patient centered care. The review did not focus on interventions which involved body massage or the use of oils and aromatherapy as the use of these may be contraindicated with SJS/TEN patients. The outcomes of interest for the review were PTSD or symptoms of PTS or anxiety or depression or self-esteem or body image or pain or quality of life (Table 1).

**Table 1. PICO review question and search terms.**

| | |
|---|---|
| **Population 1** | Adults (persons aged 18 and over) AND Burn patient or Burns or Burn injur* or Burn patient* or burns AND |
| **Population 2** | Adults (persons aged 18 and over) diagnosed with Stevens-Johnson Syndrome or Epidermal Necrolysis or toxic or Stevens Johnson syndrome or Toxic Epidermal Necrolyses or Toxic Epidermal Necrolysis or TEN or Lyell's Syndrome or Lyell Syndrome or Lyell's Syndromes or Lyell's disease |
| **Intervention** | Psychotherapy or Psycholo* AND (care or intervention* or strateg* or manag* or technique* or approach or approaches or support or nursing or accompaniment or accompagnement or treatment or treating) or Psychotherap* AND (care or intervention* or strateg* or manag* or technique* or approach or approaches or support or nursing or accompaniment or accompagnement or treatment or treating) or counselling or counsellor or Patient centered nursing or Patient centred nursing or Patient centred care or Patient centered care or Holistic nursing or affirmative therapy or cognitive therapy or anxiety AND (manag* or intervention* or support or technique*) or emotion focused therapy or PTSD AND (manag* or intervention* or support or technique*) or Post traumatic stress disorder AND (manag* or intervention* or support or technique*) or Emotional support or Self-esteem AND (manag* or intervention* or support or technique*) or Body image AND (manag* or intervention* or support or technique*) or Panic AND (manag* or intervention* or support or technique*) or Stress And (manag* or intervention* or support or technique*) or Hypnosis or Distraction techniques or Meditation or Mindfulness or Diary writing or Fear AND (manag* or intervention* or support or technique*) or Premorbid psychopathology AND (manag* or intervention* or support or technique*) |
| **Context** | Intensive care unit or Critical care or Critical care nursing or Burn units or Acute phase or Short term or A&E or Critical care or Intensive care or ICU or ICUs or ITU or ITUs or Intensive Care Unit or Intensive Therapy Unit or Intensive Therapy Units or High dependency unit or HDU or Burn Center or Burn Centers or Burn Centre or Burn Centres or Burns Unit or Burns Units or Inpatient* |
| **Outcome** | PTSD or post-traumatic stress symptoms or anxiety or depression or self-esteem or body image or pain or quality of life |

## Formation of focused question

This review was guided by the integrative review process developed by Whittemore and Knafl [30] and the PRISMA guidelines [31]. (S1 Checklist). This process allows for the inclusion of disparate methodology studies. The original review question arose from clinical practice (S.R, B.R, P.O'R, S.W, S.I-H.O, C.B). It is envisaged these findings elicited from the review may in part inform clinical practice. The PICO framework was used to develop and refine the review question. Two populations were included: firstly, adults over 18 years of age presenting with burns (Population. 1), and secondly adults with SJS/TEN (Population. 2). The search terms included synonyms and Medical Sub-Headings (MeSH) describing burns, SJS, TEN and psychotherapeutic interventions. Psychotherapeutic interventions used to manage symptoms of PTS, anxiety, depression, self-esteem, body image, quality of life or pain, within an acute care setting, were the focus (Table 1). The review question was 'What is the evidence on the psychotherapeutic interventions which have been used with either adult burns patients or patients with SJS/TEN, during the acute stage of the illness, to reduce symptoms aligned to PTS and improve quality of life?'.

## Search for the best available evidence

Both quantitative and qualitative studies were included in the review so as to capture the objective and subjective nature of the review question. The protocol for the review was published with the International Prospective Register of Systematic Reviews (CRD42020159134) [32]. A systematic search following PRISMA guidelines [31] was conducted for both populations. Two members of the review team were information specialists (Librarians; I.D; L.D) and they led the search process. The search included both French and English language papers. French papers were included as there is a national reference center for Toxic Bullous Dermatoses in

France. To test for accuracy and sensitivity of the search terms a preliminary search was conducted on MEDLINE (S1 Table). Nine electronic international databases for publications were searched including CINAHL Complete, Medline (Ovid), Embase, Scopus, Web of Science, Cochrane Library, Medline (EBSCO) PsychINFO and Medline (Pubmed). In addition, grey literature was searched using Cogprints, Grey literature reports, LENUS, Mednar, Olaster, OpenGrey, PROSPERO, Science. Gov, WHO Global Index Medicus, HAL-OAR for French research community and DUMAS. The search timeline was January 2008 to January 2021. Records were exported to EndNote and all duplicates and studies outside of the date range were removed. The search yields and exclusions are outlined, for both populations in Figs 1 and 2.

## Quality assessment

The Critical Appraisal Skills Programme (CASP) [33] was used to evaluate the methodological limitations of the included studies. Two authors (POR, PM) independently appraised the quality of the studies. Both the Randomised Control Trial (RCT) and qualitative format of the CASP were used. Not all studies were RCTs however, the RCT CASP version was the most relevant to use with quasi experimental and intervention studies (S2 Table). Papers were not excluded based on the outcome of the CASP assessment.

## Data extraction and synthesis of results

Data were abstracted from the 17 included studies (Tables 2 and 3). As the studies were heterogeneous in terms of design and outcome measures, the data were analyzed inductively using the different stages of content analysis [34].

The results and/or findings of the included studies were coded within the context of the review question, resulting in codes, sub-categories, categories and themes (Tables 4 and 5). The analysis focused on the manifest content or the obvious components of the data [35]. To ensure rigour and trustworthiness, each stage of the analysis process was repeated three times. Initially two researchers carried out the analysis process separately and then compared results until a consensus was reached. All team members agreed with the final outcome of the analysis process. The review question guided all stages of the content analysis process. The first stage, "decontextualisation", involved the development of meaning units and codes. A meaning unit is the smallest unit and comprises a combination of relevant sentences or words that relate to the review question [36]. Similar meaning units were combined into relevant codes. During the second stage, "recontextualization", all data from the studies were reread and compared to the meaning units to ensure that no relevant material was omitted. The third stage of content analysis, "categorization", involved the condensing of the meaning units and codes into sub-categories, categories and themes. According to Krippendorf [34], the theme answers the question how and the category answers the question what. Two themes were finally constructed from the content analysis process.

## Results

### Study characteristics

For the burns evidence (Population. 1) 1,062 titles were sourced in searching nine databases in addition to other sources (Fig 1). For the SJS /TEN evidence (Population. 2), 125 titles were obtained using the same databases (Fig 2). Following the removal of duplicates, 548 records remained for Population 1 and 95 records for Population 2. Six authors screened the titles and abstracts (P.O'R.,P.M.,B.W.,C.K.,S.H.,A.C) with A.C acting as an arbitrator, when required.

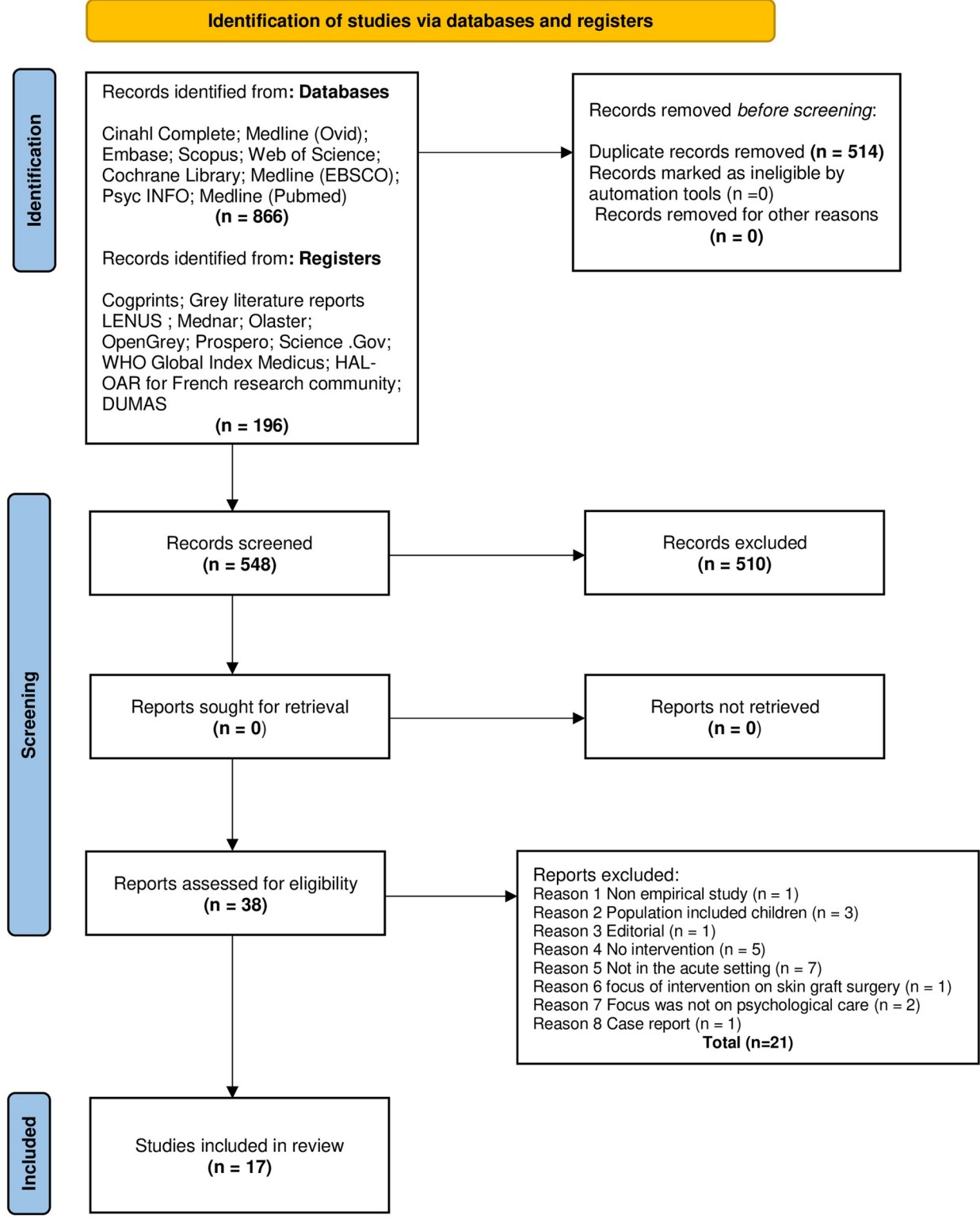

**Fig 1. PRISMA flow diagram for burns evidence (Population 1)** [31].

**Fig 2. PRISMA flow diagram for SJS/TEN evidence (Population 2)** [31]**.**

**Table 2. Characteristics of the included quantitative studies.**

| Author/Country | Aim/Purpose | Study Design | Study Participants | Outcome Measures | Method of Analyses | Key Findings |
|---|---|---|---|---|---|---|
| Delfani et al. [46] Iran | Compare the effects of muscle relaxation and mental imagery techniques on pain intensity in in patients with second degree burns | Quasi-experimental study | n = 135 (all male). Age range 20–45 years | VAS | One-way analysis of variance (ANOVA), Chi-square test and one sample Kolmogorov-Smirnov test | Both interventions viz. muscle relaxation and mental imagery significantly reduced pain intensity (p = 0.02; p < 0.01) in patients with a second-degree burn wound. |
| Fauerbach et al. [37] USA | Evaluate the feasibility of conducting an RCT of the SMART (Safety, Meaning, Activation, and Resilience Training) intervention vs nondirective supportive psychotherapy (NDSP) in a sample of acutely hospitalized adult survivors of burn injury. | A proof-of-concept, parallel group RCT | n = 50 but due to drop out and missing data final sample size was n = 40. Age range 18–62 years | DTS ASD PHQ-9 MCSQ | Descriptive statistics (range, median, SE) | The findings suggest that (1) it is feasible to conduct an RCT of brief CBT (i.e., SMART) vs NDSP in an acutely injured, hospitalized sample of survivors of burn injury and (2) brief CBT has the potential to yield clinically significant outcomes in this population. |
| Jafarizadeh et al. [38] Iran | Compare the effectiveness of hypnosis and 'neutral hypnosis' (as a placebo in the control group) in decreasing background burn pain | Quantitative— blinded, randomised, placebo-controlled study. | n = 60 (all male). Mean age 30.5±9.11 years | VAS PHQ-9 BSPAS SF-MPQ The Stanford hypnotic clinical scale | Descriptive statistics (mean ± standard deviation and frequency (%)), Kolmogorov-Smirnov test to identify the normal distribution of data, independent t-test, Chi-square test and Fisher's exact test, Repeated measures ANOVA | No significant difference between groups in reduction in background pain intensity. Significant reduction in background pain quality and pain anxiety in the intervention group during the four hypnosis sessions (p<0.0005). |
| Berger et al. [50] Switzerland | Measure the influence of a new pain management including hypnosis in a critical care setting on pain intensity and the patients' anticipation of pain before treatment. | Intervention study with matched controls | n = 23 + n = 23 historical controls. Age 36±14 years old. | VAS ESAS | Data provided as mean ±S. D., median and range. Comparison of baseline continuous variables between groups with one-way ANOVA, and non-parametric variables with x2 tests, or Wilcoxon test. Two-way ANOVA used to analyse evolution of opioid dose delivery over time. | The pain protocol using hypnosis resulted in the early delivery of higher opioid doses/24 h (p<0.0001) followed by a later reduction with lower pain scores (p<0.0001), less procedural related anxiety, less procedures under anaesthesia, reduced total grafting requirements (p = 0.014), and lower hospital costs per patient. |
| Najafi Ghezeljeh et al. [39] Iran | Evaluate the effects of massage and music on pain intensity, anxiety intensity and relaxation level in burn patients. | Randomized controlled clinical trial with factorial design | n = 240 divided into 4 following groups (i) control group (n = 60), (ii) massage group (n = 60), (iii) music group (n = 60) and (iv) music-plus-massage group (n = 60). Mean [standard deviation (SD)] age of the participants was 32.23 (8.43) years. | VAS | One-way analysis of variance (ANOVA), Scheffe ad hoc test and Chi-square test were applied. Mean scores of changes in variables before and after intervention were considered to compare groups. Within group comparison (before and after intervention), paired t-test and Chi-square test were used. | There was a significant difference in the mean change scores of pain intensity between the control group and music group (p < .001), massage group (p < .001) and the music plus massage group (p < .001). |

*(Continued)*

**Table 2.** (Continued)

| Author/Country | Aim/Purpose | Study Design | Study Participants | Outcome Measures | Method of Analyses | Key Findings |
|---|---|---|---|---|---|---|
| Mohammedi Fakhar et al. [40] Iran | Determine the effect of jaw relaxation on pain anxiety related to dressing changes in burn injuries. | A randomised clinical trial with a control group. | N = 100 (72 male, 28 female) Mean age 32.95 years (SD = 11.33) ranging from 18 to 60 years. | BSPAS | Descriptive statistics, chi-square test, dependent and independent t-test and Fisher's exact test | Following jaw relaxation intervention (before dressing) there was a significant difference in the experimental group (p<0.05). Post-dressing pain anxiety of the experimental group was less than the control group (p<0.05). However, there was no significant difference between before and after dressing pain anxiety (after intervention) in the experimental group (p = 0.303). |
| Ozdemir & Sarritas [47] Turkey | Determine the effect of yoga on self-esteem and body image of burn patients. | A quasi-experimental pre-test post-test with a control groups clinical trial. | N = 110 (52 male, 58 female) Age range 18–85 years. | The Body Image Scale RSES | Chi-square test for comparison of percentage, mean, standard deviation, and control variables. Independent sample $t$-test for intergroup self-esteem and body image mean score comparison. Paired sample test for in-group self-esteem, a body image mean score comparison was used after yoga practice, and Cronbach's alpha for reliability analysis. | After yoga practice, there was a statistically significant increase and improvement in the self-esteem (p < 0.05) and body image of the experimental group (p < 0.05). A statistically significant decline in the score average of pre-test and post-test of body image of the patients in the control group was observed (p<0.05). |
| Li et al. [45] China | Observe the effect of a rehabilitation intervention on the comprehensive health status of patients with hand burns. | Randomized controlled design | n = 60 (n = 30 intervention group + n = 30 control group) (47 male, 13 female) Mean age±SD for control group 38.33 ±14.10 and intervention group 35.5±12.59. | BSHS-B | Descriptive statistical analysis used to determine means, ranges, and standard deviations of the variables, Student's t-test used to compare comprehensive health levels and four sub-domains at the baseline between groups, and analysis of variance (ANOVA) was also used to determine whether there was an effect of intervention on comprehensive health level. | The rehabilitation intervention group had significantly better scores than the control group for comprehensive health (p<0.001), physical function (p<0.001), psychological function (p<0.001), social function (p<0.001) and general health (p<0.001). |
| Morris et al. [41] South Africa | Ascertain the feasibility and potential effect of a Virtual Reality system used in conjunction with analgesia, on reducing pain and anxiety in adult burn patients undergoing physiotherapy treatment, compared to analgesia alone. | A randomized (condition only), single-blind (assessor blinded only), single-subject, pre–post experimental case series (within-subject) design | n = 11 (3 female, 8 male) Median age 33 years (range 23–54 years). | NPRS BSPAS | Box-and-whisker plot method, Chi-square tests as well as the Student's paired t-test were used to analyse data. | A marginal (p = 0.06) to insignificant (p = 0.13) difference between the two sessions (analgesia with VR and analgesia without VR) in reducing pain was found. No significant difference (p = 0.58) was found between the two sessions (analgesia with VR and analgesia without VR) for anxiety. |

*(Continued)*

**Table 2.** (Continued)

| Author/Country | Aim/Purpose | Study Design | Study Participants | Outcome Measures | Method of Analyses | Key Findings |
|---|---|---|---|---|---|---|
| Park et al. [48] South Korea | Evaluate the effects of Relaxation Breathing on procedural pain and anxiety during burn dressing changes. | A quasi-experimental, pre-test-post-test comparison group design without random assignment to groups. | N = 60 (n = 30 experimental group and n = 30 control group) (29 male, 31 female) | VAS | Descriptive statistics, including mean, median, and standard deviation (SD), were obtained to describe the sociodemographic and burn-specific variables. The homogeneity test was used to detect any significant group differences in the demographic data and pre-test measures. | The pain scores significantly differed between the 2 groups after intervention (RB group vs. control group, P = .01) and over time (pretest vs. posttest, P = .001). The anxiety scores significantly differed between the 2 groups (P = .01) and over time (P = .02). |
| Fauerbach et al. [42] USA | Determine if contradictory coping messages would lead to an approach–avoidance coping conflict and to determine if experiment-induced coping conflict is also associated with higher distress. | Randomised, within-subject crossover design | n = 59 (45 male, 14 female) | IES | Analyses were conducted to test for pre-test differences using analysis of variance (ANOVA), Pearson's w2 statistic, or Fisher's exact test, as appropriate | Participants in the process-then-suppress condition, relative to the suppress-then-process condition, were significantly more likely to exhibit approach–avoidance coping conflict. Approach–avoidance coping conflict was associated with greater re-experiencing symptoms. The order of coping skill training can influence treatment outcome, success of coping methods, and overall levels of distress. |
| Wiechman et al. [43] USA | Determine the effects of hypnosis on postburn itch and pain relief. | A andomized control trial with a control group. | N = 27 (62% Caucasian & 60% male) | NPRS 5-D Itch Scale | Not outlined | There were no significant differences between the groups on any outcome measure and both groups demonstrated improved pain and itch over time. There was a large effect size for Itch as measured by the NRS (intensity) and the 5D Itch Scale from baseline to 1 month. |
| Seyedoshohadaee et al. [49] Iran | Determine the effect of a short-term training course by nurses on body image in patients with burn injuries. | A semi-experimental single-group survey | n = 130 (65 women and 65 men) | SWAP | Descriptive statistical analysis used to determine means, paired sample t-test, used to outline the difference between the mean scores of body image before and after educational interventions | The mean scores of the body image of patients before and after the intervention were 49.44 ±11.39 and 41.63±11.89, respectively. There was a significant difference between the mean scores of body image before and after educational interventions (T = 6.013, P≤0.001). |

(*Continued*)

**Table 2.** (Continued)

| Author/Country | Aim/Purpose | Study Design | Study Participants | Outcome Measures | Method of Analyses | Key Findings |
|---|---|---|---|---|---|---|
| Mamashli et al. [44] Iran | Determine the effect of implementing interventions based on mental empowerment through multimedia education in burn patients | A randomized clinical trial with a control group | n = 50 control group; n = 50 intervention group (44% female and 56% male) | BSHS-B | Descriptive and inferential statistics (Chi-square and independent and paired t tests for the distribution of normal variables), Fisher's exact test, nonparametric tests e.g., Mann-Whitney, Wilcoxon and Friedman test and Dunn test, with Bonferroni's correction, Spearman correlation coefficient | Before interventions, the mean of mental dimension in intervention and control groups were 2.08±0.59 and 1.64±0.47, respectively (p<0.001). Three and six months after the Intervention, they were 3.37±0.93 and 2.24±0.4, 4.11±0.74 and 2.75±0.58, respectively (p<0.001). |
| Pruskowski et al. [51] USA | Determine the impact of a therapy dog programme | Cross sectional study | n = 14 patients and n = 23 staff | None used | Not outlined | Most patients reported improved pain and anxiety after working with the therapy dogs. |

The remaining team worked in pairs with B.W working with both C.K and P.M. Regular review meetings were held to agree on outcomes. 38 full text records for Population 1 and 3 for Population. 2 were reviewed, whilst adhering to the inclusion and exclusion criteria. Through the screening process 17 studies were included in the review. No studies relating to the psychotherapeutic interventions used with SJS/TEN patients were found. All included studies related to psychotherapeutic interventions used with adult burns patients. Fifteen studies were quantitative and included RCTs (n = 9) [37–45] quasi-experimental study (n = 4)

**Table 3. Characteristics of the included qualitative studies.**

| Author/Country | Aim/Purpose | Study Design | Study Participants | Method of Analyses | Key Findings |
|---|---|---|---|---|---|
| Badger and Royse [53] USA | Explore burns survivors' descriptions of compassionate health care to explicate and better understand this concept within the context of burn care | Qualitative focus group interviews | n = 31 (18 female, 13 male). Age range 23–70 years | Qualitative thematic data analysis | Three primary themes identified with subthemes. 1) respect the person (subthemes: establishing an empathic connection, restoring control through choice, providing individualized care, and going above and beyond), 2) communication (subthemes: interpersonal and informational), and 3) provision of competent care. The three primary themes were components of compassionate care; it was not defined by a single characteristic, behaviour, or skill but might be best understood as the convergence of the three themes. |
| Kornhaber et al. [52] Australia | Explore burn survivors' experiences of providing and receiving inpatient peer support to develop an in-depth understanding of the influence during the rehabilitation journey. | A descriptive phenomenological methodology. | n = 21 (20 male, 1 female). Average age 44 years ranging from 21–65 years. | Transcripts analysed using Colaizzi's seven steps method of data analysis. | Inpatient peer support had both a positive and negative impact on the rehabilitation journey. Findings demonstrated that peer support assisted with fostering reassurance, hope and motivation in burn rehabilitation. Inappropriate matching of peer supporters and bad timing in providing the support could impact negatively on participants. |

**Table 4. Theme 1 empirically supported psychotherapeutic treatments.**

| Theme 1 | Empirically supported psychotherapeutic treatments | | |
|---|---|---|---|
| **Category** | 1. Relaxation Therapy | 2. Hypnosis | 3. Cognitive behavioural therapy; Virtual reality |
| **Sub-category** | These techniques reduce burn patients' pain. Mental imagery had more reducing effect on pain intensity. A simple and inexpensive method of jaw relaxation can reduce the pain anxiety related to dressing in patients with burns. Decrease in distress and increase in relaxation levels during dressing changes. | Hypnosis for reduction of background pain and pain anxiety in patients with burns Hypnosis as part of a pain protocol (carried out by a qualified hypnotherapist) Hypnosis to treat burn pain and non-burn-related pruritus | CBT—Safety, Meaning, Activation, and Resilience Training (SMART) protocol Low-cost Virtual Reality (distraction) system used in conjunction with pharmacological analgesia |
| **Codes** | A significant difference in the mean score of pain intensity after dressing on the second day of burn between the control and the relaxation groups and mental imagery groups. A significant difference between the mean post-dressing pain anxiety scores in the experimental and control groups. Significant difference in pain and anxiety scores between two groups | Hypnosis is generally effective for the treatment of pain and pain anxiety, which is consistent with many studies confirming the effect not only on burns but also on many other types of acute or chronic diseases The study shows that a protocol in pain management including hypnosis reduced patient anxiety and exposure to pain, increased early opioid delivery, and decreased general anaesthesia requirements, hospital length of stay and costs Itch and pain seem to improve over time. Barriers to making hypnosis a feasible nonpharmacological option for patients who report burn pain and itch as impacting quality of life should be addressed. | CBT can target maladaptive ruminations, a core cognitive component of depression. No significant difference was found between the two conditions (with or without VR administration) in reducing anxiety. |
| **Reference** | [40, 46, 48] | [38, 43, 50] | [37, 41] |

[46–49] intervention study (n = 1) [50]; and a cross sectional study (n = 1) [51]. Two qualitative studies, a descriptive phenomenological study (n = 1) [52] and a qualitative descriptive (n = 1) [53], were included. In total 1,244 patients were included in the studies. Six of the

**Table 5. Theme 2 alternative psychotherapeutic treatments.**

| Theme 2 | Alternative psychotherapeutic treatments | |
|---|---|---|
| Category | 1. Training programmes | 2. Therapeutic Relationships |
| Sub-category | Music & Massage Yoga Nidra Rehabilitation intervention1 Coping skills training Training Course on body image Multimedia Psychosocial empowerment intervention Therapy Dog programme | Compassionate care Inpatient Peer support for burn survivors |
| Codes | Improvement in pain and anxiety intensity with increased levels of relaxation Improvement in self-esteem and body image The improvements of the rehabilitation intervention model were significantly greater than those observed in the control group for comprehensive health and the four sub-domains Coping skills training is important viz. firstly teach (suppression) (training in stabilizing and calming methods) then secondly (processing) (mindfulness, counselling, exercise The effects of short-term training courses by nurses on the body image in patients with burn injuries Psychosocial empowerment interventions through multimedia education in burn patients. A therapy dog program with the intent of improving duration and quality of rehabilitation sessions and physical therapy | 1. Empathic connection—attentive listeners, seeking to understand the survivors' experiences and concerns, seeing someone as a human being and try and understand what their experience could 2. Interpersonal communication—helps to validate that patient exist as individuals 3. Provision of Competent Care—demonstration of competence during patient-care-giver interactions and treatment Inpatient peer support had both a positive and negative impact on their rehabilitation journey. • Encouragement, inspiration and hope • Reassurance • The importance of timing • The same skin • Appropriate matching |
| Reference | [39, 42, 44, 45, 47, 49, 51] | [52, 53] |

studies were carried out in Iran [38–40, 44, 46, 49], five in the United States of America [37, 42, 43, 51, 53] and one study each in Switzerland [50], Australia [52], Turkey [47], China [45], South Africa [41], and South Korea [48]. The psychotherapeutic interventions included in the studies were relaxation techniques [40, 46, 48], hypnosis [38, 43, 50], cognitive behavioural therapy (CBT) [37], virtual reality [41], psychosocial empowerment intervention [44], rehabilitation intervention [45], dog therapy programme [51], yoga [54], body image training course [49], music and massage [39], coping skills training [42], compassionate care [53] and peer support [52]. Sixteen outcome assessment tools were included in the 15 quantitative studies. Pain intensity was measured with three assessment tools, the visual analog scale (VAS) [55], in four studies [38, 46, 48, 50]; the Numeric Pain Rating Scale (NPRS) [56], in two studies [41, 43]; and the Edmonton Symptom Assessment Scale (ESAS) [57] in one study [50]. Pain anxiety was measured using two tools, the VAS-Anxiety [58] in two papers [39, 48] and the Burn Specific Pain Anxiety Scale (BSPAS) [59] in three studies [38, 40, 41]. Acute stress disorder was measured using the Acute Stress Disorder Scale (ASD) [60] in Fauerbach et al. [42]. Pain quality was measured with the short form-McGill pain questionnaire (SF-MPQ) [61] in Jafari-zadeh et al. [38]. Symptoms of PTS were measured using three assessment tools viz. the Impact of Events Scale (IES) [62], in Fauerbach et al. [42] the Davidson Trauma Scale (DTS) [63, 64], in one study [37] and the ESAS in Berger et al. [50]. Depression was measured using the Patient Health Questionnaire -9 (PHQ-9) [65] in one study [37] and the ESAS in another [50]. Body image was measured with The Body Image Scale [66], and the Satisfaction with Appearance scale [67] in Ozdemir and Saritas [47] and Seyedoshohadaee et al. [49] respectively. The Rosenberg Self Esteem Scale) [68] was used to measure self-esteem levels [47]. The quality of life assessment tools that were used were the Quality of Life [BHS-B (Burn Specific Health Scale) (QoLBSHS-B) [69], in two studies [44, 45] and the ESAS in another [50]. The Modified Coping Strategy Questionnaire (MCSQ) [70] was used to measure the frequency of coping skills in Fauerbach et al. [42]. Hypnotisability was measured with the Stanford Hypnotic Clinical Scale [71] in one study [38]. An itch scale, the 5-D Itch scale [72] was used in Wiechman et al. [43]. In relation to the two qualitative studies, Badger and Royse [53] aimed to explore patients' descriptions of compassionate care so as to better understand the concept as applied within the context of caring for burns patients. The second qualitative study, Kornhaber et al. [52] investigated burn survivors' experiences of either giving or receiving peer support while in hospital. The publication years of studies ranged from 2009 to 2020. In five of the studies the patients were not randomly assigned to the treatments [46, 48–51]. Moreover, patients and health care personnel were not 'blind' to treatment in ten studies [40–43, 46–51]. The method of analysis was not outlined clearly in two of the papers [43, 51].

## Themes

Through content analysis, two themes were identified, viz. (i) Empirically supported psychotherapeutic treatments, (ii) Alternative psychotherapeutic treatments. The themes reflected the types of psychotherapeutic interventions included in the review. The first theme included three categories viz. relaxation therapy, hypnosis and cognitive behavioural therapy, virtual reality. The second theme comprised two categories viz. training programmes and therapeutic relationships.

### Theme 1: Empirically supported psychotherapeutic treatments

**Category 1: Relaxation therapy.** Data from three quantitative papers reflected this category [40, 46, 48]. The category focused on how relaxation techniques may impact pain intensity [46], and pain anxiety intensity levels in burns patients [40, 48]. Delfani et al. [46] used the

VAS as an outcome measure for pain intensity. Pain anxiety was measured using the BSPAS by Mohammadi Fakhar et al. [40] and the VAS- Anxiety measure by Park et al. [48].

In relation to pain intensity, Delfani et al. [46] in a quasi-experimental study using the VAS established that the use of muscle relaxation and mental imagery helped to reduce the pain experienced by patients. The study comprised 135 patients who were assigned into two experimental groups (muscle relaxation and mental imagery) and a control group. On the second day of burn injury, post dressing, a significant difference was observed in the mean score of pain intensity between the control and the relaxation groups (P = 0.02) and between the control and the mental imagery groups (P<0.01). Confounding variables may not have been accounted for due to the lack of randomization of patients to groups. The authors suggested that muscle relaxation and mental imagery may be used in combination with medication to reduce pain in patients.

Mohammadi Fakhar et al. [40] found that, by using a simple and inexpensive method of jaw relaxation with patients (n = 100) there was a significant reduction in pain anxiety in the experimental group. The BSPAS was used to measure pain anxiety before and after dressings. After the jaw relaxation intervention and before the burns dressing, using a dependent t-test, a significant difference (P = .000) in pain anxiety was observed between the experimental (mean = 42.56, SD+- 21.98) and control group (mean = 51.10, SD+- 19.90). With regard to post dressing, the independent t-test showed less pain anxiety in the experimental group (mean = 44.77, SD+- 23.06) compared to the control group (P = 0.048) (mean = 53.54, SD+- 20.67) [40]. Whilst further research is required, there is a possibility of reducing patients' pain anxiety by teaching them how to use this simple relaxation technique.

A quasi-experimental study which aimed to establish the effects of relaxation breathing, used by burns patients (n = 60) during dressing times, on procedural pain and pain anxiety found a significant difference in pain scores between the intervention group (n = 30) and control group (n = 30), (p = .001) and over time, (p = .001) [48]. In addition to the decreased pain level in the intervention group, Park et al. [48] observed a greater decrease in anxiety levels, as measured by the VAS-Anxiety, between both groups (p = .01). Relaxation breathing techniques is an intervention to be considered for pain and anxiety relief in the care of burns patients.

**Category 2: Hypnosis.** The use of hypnosis in burns patients to reduce pain and anxiety was reflected in three quantitative research papers [38, 43, 50]. The outcomes measured and the assessment tools used were as follows; Jafarizadeh et al. [38] and Berger et al. [50] assessed pain intensity with the VAS and Wiechman et al. [43] used the NPRS. Pain anxiety was assessed by Jafarizadeh et al. [38] using the BSPAS. Pain quality was measured using the SF-MPQ by Jafarizadeh et al. [38]. The ESAS was used by Berger et al. [50] to assess patient related symptoms such as depression, anxiety and well-being.

In an RCT, where hypnosis was used to reduce pain intensity, background pain quality and pain anxiety, Jafarizadeh et al. [38] reported significant findings in relation to pain quality and pain anxiety but not pain intensity. In the study, 60 men with burns were included. The SF-MPQ was used to measure pain quality. Following four hypnosis sessions, a significant reduction in the mean scores of the intervention group was observed ($F (1.628, 47.219) = 47.356$, p<0.00005). In relation to pain anxiety, which was measured with the BSPAS the intervention group presented with a reduction in mean scores following all sessions ($F (2, 58) = 35.215$, p<0.00005) [38]. Although the findings indicate that hypnosis may be used to reduce pain and pain anxiety, further research would be required on different populations.

An intervention study by Berger et al. [50] also employing hypnosis as part of a pain protocol, found a reduction in pain intensity in the intervention group of burns patients (n = 23). The VAS [55] daily mean scores was significantly reduced from 1.4±1.7 to 0.9±1.3 points

(p < 0.0001). In addition, both anxiety and depression were reduced as measured by the ESAS from 3.2±2.9 to 1.2±1.7 points (p<0.0001) and 1.8±2.5 to 1.0±1.6 points (p<0.014) respectively [50]. However, Wiechman et al. [43] in an RCT found no significant differences between intervention and control groups for pain and itch, noting that both improved with time. The analysis process was not outlined in this study. Nonetheless, the authors noted that hypnosis may be worth considering in the reduction of pain and pain anxiety in burns patients.

**Category 3: Cognitive behavioural therapy; virtual reality.** The data from two quantitative papers were included under this category [37, 41]. Symptoms of PTS were assessed by Fauerbach et al. [37] using the DTS, PHQ-9 and the ASD. Morris et al. [41] used the NPRS to measure pain intensity and measured anxiety with the BSPAS.

Fauerbach et al. [37] in a randomized proof of concept study, used a Cognitive Behavioural Therapy (CBT)-based intervention referred to as Safety, Meaning, Activation and Resilience Training (SMART) versus nondirective supportive psychotherapy (NDSP). They found that the group assigned to using the SMART intervention had substantially improved median scores for depression and symptoms of PTSD at post intervention and one month follow up stages in comparison to the NDSP group. Adapted from the work of Foa et al. [73] the SMART intervention included anxiety management training, imaginal exposure therapy, and cognitive restructuring of safety and meaning schema. Depression was measured using the PHQ-9 where the median scores for the SMART group were 5.0 post intervention and 6.0 at 1-month follow up (clinical cutoff for PHQ-9 was 10 with higher scores denoting greater severity). The median scores for the symptoms of PTSD, measured by the DTS were 16 post-intervention and 6.5 at 1-month follow up (clinical cutoff for DTS is 40 with higher scores denoting greater severity). Consequently, the proof-of-concept study highlighted the feasibility of undertaking an RCT using SMART intervention with hospitalized burns patients [37].

A study in burns patients (n = 11), involved an intervention of low-cost virtual reality (VR) in conjunction with analgesia to reduce pain and anxiety when undergoing physiotherapy treatment. The authors found no significant difference (p = 0.58) in reducing anxiety (assessed with the BSPAS) by using analgesia with or without VR [41]. In reducing pain, a marginal (p = 0.06) to insignificant difference (p = 0.58) was observed between both sessions (analgesia with VR and analgesia without VR) [41]. Pain intensity was measured using the NPRS. The results need to be considered with caution due to the small sample size.

## Theme 2: Alternative psychotherapeutic treatments

**Category 1: Training programmes.** Data from seven quantitative studies were included under this category [39, 42, 44, 45, 47, 49, 51]. In relation to outcomes and measures, Najafi Ghezeljeh et al. [39] used the VAS-Anxiety to assess pain anxiety. The IES was used to measure PTS symptoms [42]. Body image was assessed using the Body Image Scale [47]. Ozdemir and Saritas [47] measured self-esteem with the RSES. Quality of life was assessed with the Quality of Life [BHS-B Scale] [44, 45]. Fauerbach et al. [42] assessed the frequency of coping skills with the MCSQ.

A psychosocial empowerment intervention via multimedia training, used in an RCT, found a significant increase in the mean of the mental dimension as measured by the QoLBSHS-B three and six months after intervention (p<0.001). It appeared to focus on empowering patients through a self-care programme [44]. In addition, interventions such as a nurse led rehabilitation intervention programme, which included client centered therapy, helped to improve psychological function (p<0.001) and social relations (p<0.001) [45]. In a cohort of burns patients (n = 12) Pruskowski et al. [51], using a survey, reported an improvement with both pain and feelings of anxiety in a study involving a dog therapy programme. Selected burns patients were visited by therapy dogs and handlers. The analysis of the data was not clearly outlined in the

study. Similarly, a body image improvement training course showed significant increases in body image scores using the Satisfaction with Appearance Scale (SWAP) following intervention [49]. There was very little information provided on the actual intervention programmes used in these studies, therefore the findings should be considered with caution.

Levels of body image (The Body Image Scale) and self-esteem (The Rosenberg Self Esteem Scale) showed a significant difference between the experimental and control groups (p<0.05), using a yoga programme, with an increase in the experimental group [47].

Using the VAS, a reduction in pain intensity was observed in an RCT using music and massage [39]. The trial had 240 participants, divided evenly into four groups (i) control, (ii) massage, (iii) music and (iv) music plus massage. Using the Scheffe ad hoc test, there was a significant difference in the mean change scores of pain intensity, as measured by the VAS, between the control group and music group (p < .001), massage group (p < .001) and the music plus massage group (p < .001).

The order in which coping skills training is carried out can influence the re-experiencing of distressful symptoms following acute burn injury [42]. Fauerbach et al. [42] purport that adaptive coping should be taught early before less efficacious coping strategies as the earlier coping strategy, post trauma, will be used most frequently. Using experimental conditions with adults with acute burns injuries (n = 59), it was found that the group, who were using the process-suppression (approach coping) condition as opposed to the suppression-process condition (avoidance coping), was more likely to experience approach avoidance conflict which was associated with greater re-experiencing of symptoms (p<0.01). The outcome measures used were the IES and the MCSQ.

**Category 2: Therapeutic relationships.** Two qualitative papers reported the importance of relational and therapeutic support for burns patients [52, 53].

Within focus group interviews, participants (n = 23) were asked to reflect back on what compassionate care meant to them while being cared for in an acute burns unit setting [53]. Findings included the importance of being treated with respect and being seen as a person rather than an injury/disease.

> ". . .remembering that someone is a human being and trying to understand what their experience could be. I'm not just an object or a medical procedure. I'm a person.." (P775).

Patients valued an empathetic connection with carers, with being listened to, having choices and having staff who went 'above and beyond' to meet their needs:

> "They did their job, they did it fantastic, but it was just those little things they did on the side that makes it feel like they really do care" (P777).

It was important for individuals to be informed about their procedures and progress. Carers, who were competent in what they did and said, instilled confidence and a feeling of safety and security for patients [53]. Kornhaber et al. [52] also noted that inpatient peer support by burns survivors had the potential of providing current patients with encouragement and hope:

> ". . .that gave me a lot of courage and inspiration, I can do it as well, . . .able to one day get out of this and go about doing my work" (P113).

In addition, timing of the support visit was deemed important to note i.e. not during the acute phase as patients may not be ready. Ensuring that the peer supporter was an appropriate match required careful consideration [52].

## Discussion

Following the screening process, 17 studies were sourced on psychotherapeutic interventions used in the care of adult burns patients. The review revealed no studies that related specifically to SJS/TEN patients. Due to the heterogenous nature of the evidence supporting this review, a meta-analysis was not possible. In its place content analysis [34] was employed to analyse the data. Fifteen of the papers were quantitative, of which nine were RCTs, and two were qualitative. Following the analysis process two themes were developed. The first of these, namely empirically supported psychotherapeutic treatments, reflected data from eight research papers. There were three categories under this theme. The first category relaxation therapy focused on the effects of relaxation techniques on pain intensity and pain anxiety. The studies showed some significant results in relation to the reduction of pain intensity/pain anxiety in burns patients by using relaxation techniques, some of which may be considered using with SJS/TEN patients [40, 46, 48]. The second category hypnosis included three studies and highlighted some significant findings in relation to pain quality [38], pain anxiety [38], pain intensity [43, 50], anxiety [50] and depression [50]. However, one study found no significant differences for pain and itch. In fact, it noted that both improved over time [43]. As most of these studies focused on pain as an outcome, it would be important to carry out further research, particularly with SJS/TEN patients, to evaluate the effects of relaxation techniques and hypnosis on outcomes such as the symptoms of PTS.

Category three CBT; VR included two quantitative studies. Outcomes measured in the studies included symptoms of PTS [37, 41], and pain intensity [41]. Fauerbach et al. [37] reported the use of the SMART intervention, which included the use of anxiety management training, imaginal exposure therapy, and cognitive restructuring of safety and meaning schema. With further research, the use of the SMART intervention may be worth considering to reduce symptoms of PTS with burns patients.

A study, using VR with or without analgesia, involved a small number of participants and there were no significant findings for reduced anxiety levels, whilst there was a marginal to insignificant effect for reducing pain [41]. Consequently the results need to be considered cautiously.

The second theme alternative psychotherapeutic treatments included data from nine studies. There were two categories under this theme. The first category was training programmes. The outcomes measured were pain anxiety, symptoms of PTS, body image, self-esteem, quality of life and frequency of coping skills.

Although interventions such as the psychosocial empowerment [44], nurse led rehabilitation [45], dog therapy [51] and a body image improvement [49] outlined improvements in pain, anxiety, psychosocial function and body image there was very little information provided about the details of such interventions consequently, the results require cautious consideration. The use of yoga showed significant results in both body image and self-esteem levels [47]. The evidence outlines the possibility of considering the use of yoga as part of a nurse led intervention to improve the self-esteem and body image of burns patients.

Fauerbach et al. [42] highlighted the importance of considering the use of anxiety management and relaxation techniques (suppression techniques–avoidance coping) as the first line psychological management of patients, during the acute post trauma stage of burns. This can help to recreate a sense of calmness and control for the patient, rather than using exposure and cognitive restructuring (process-approach coping) as first line interventions. This finding could be taking into consideration when deciding on the staging of interventions. The second category included two qualitative studies, the importance of compassionate care [53] and the potential of using peer support [52]. These studies reported the importance of communication, empathetic connection and the provision of hope and encouragement to patients.

The majority of the studies included focused on levels of pain and pain anxiety as outcomes. More research is required to address the impact of interventions on symptoms aligned to PTS.

Whilst there is a dearth of studies relating to the psychological impact of SJS/TEN on patients' lives, the existing studies acknowledges the range of psychological effects on patients' well-being such as anxiety, depression and PTSD [14–19, 21, 22]. The psychological sequelae of burns are well documented. High prevalence rates of anxiety and depression are observed in patients, both whilst in hospital and following discharge [27].

Hefez et al. [21] recommends the need for a multidisciplinary approach in the prevention and management of PTS symptoms in SJS/TEN patients. They highlight the need for psychiatric assessment of patients in the acute stage as well as during follow up care and for the examination of other PTSD preventive strategies that may be put in place for such patients. The study showed that the main risk factor for subsequent PTSD, at 6 months post-acute phase of SJS-TEN, was the underlying psychological fragility of the patient. This important finding emphasizes the need for preventive strategies in the already fragile patients, from the acute phase of the disease [21]. In comparison to Theme I, the studies under Theme 2 focused more on the long-term recovery of patients [42, 44, 47, 49]. However, only one of the studies assessed the long-term outcomes of the intervention at three and six months after intervention [44]. The qualitative studies under Theme 2 endeavored to establish firstly, the caring behaviours that made a difference to patient recovery and secondly, how inpatient peer support impacted the rehabilitation journey of the patient. It is important to use longitudinal studies to address the long-term impact of conditions such as burns and SJS/TEN on patients' lives. In relation to SJS/TEN, due to the rarity of the conditions few prospective studies and RCTs are carried out in many of the adjunctive therapies [29]. Shanbhag et al. [23] similarly highlights that research on psychological care treatments for SJS/TEN is lacking. Evidence suggests that these conditions may have long lasting implications on the quality of life of those affected. Lee et al. [13] purports the need for clinicians to appreciate the chronic phase of SJS/TEN. Consequently, more research is required to address these anomalies so as to strengthen treatment guidelines and care pathways.

A recent systematic review, addressing the evidence for psychological therapies in treating adults with PTSD, found trauma and non-trauma focused CBT to be beneficial, with trauma focused being more effective [74]. This approach is in line with the SMART intervention in a proof-of-concept study as outlined by Fauerbach et al. [37] Therapies, such as yoga, mindfulness-based stress reduction, hypnosis and meditation, have been supported by some studies. However, there is a lack of well-designed trials [75]. Similarly, in a systematic review, on non-pharmacological interventions for anxiety in burn patient, a recommendation was made for more well designed RCTs in the area [76]. Unlike this review Fardin et al. [76] included studies on massage and aromatherapy. The stress reduction interventions for burns patients in our review, notably relaxation techniques [39, 40, 46, 48] and hypnosis [38, 43, 50], may be worth considering to explore in SJS/TEN patients along with further testing.

The care environment and caring relationships are both central and critical to caring for SJS/TEN patients. In line with Badger and Royse [53], O'Reilly et al. [6] highlighted that patients and their families valued care that was compassionate, dignified and respectful. It was important to both patients and their families to have a 'liaison' person, who was calm, confident and well prepared, so as to communicate from both an informational and emotional perspective [6]. The burns/dermatology specialist nurse, experienced in the needs of the SJS/TEN patient, is in one sense the conductor helping to coordinate the input of the other specialties, and is also the person who is responsible for the administration of physical treatments directly with the patient, such as dressings. This primary 'hands-on' relationship is central and is critical in helping to allay feelings of anxiety and stress for the patient.

A strength of this review is the use of a rigorous approach to the comprehensive and in-depth search strategy process. However, the lack of studies or evidence on psychotherapeutic interventions for adult SJS/TEN patients is a limitation. However, it does signal the importance of a need for research in this area. The 17 studies relate to interventions used with adult burns patients only. The heterogeneous nature of the included evidence included has meant that meta-analysis could not be undertaken. In addition, patients were not randomly assigned in five of the studies and, in ten of the studies, patients or healthcare personnel were not blind to treatment. Therefore, there was potential for selection and allocation bias. Due to the disparate nature of the studies drawing conclusions was difficult.

## Conclusion

In conclusion, hospitalized adult patients with SJS/TEN are vulnerable to psychological sequelae. It is evident that more research is required to rectify the dearth of evidence in this area and to establish psychological care pathways, specific to the needs of such patients. There is a possibility that, following further research, some of the interventions deployed in burns patients may be applicable to SJS/TEN patients, particularly stress reduction techniques such as anxiety management, meditation, and relaxation.

## Supporting information

**S1 Checklist. PRISMA checklist.**
(DOCX)

**S1 Table. Medline search.**
(DOCX)

**S2 Table. CASP.**
(XLSX)

## Author Contributions

**Conceptualization:** Pauline O'Reilly, Barbara Whelan, Bart Ramsay, Sarah Walsh, Saskia Ingen-Housz-Oro, Christopher B. Bunker, Isabelle Delaunois, Liz Dore, Siobhan Howard, Sheila Ryan.

**Data curation:** Pauline O'Reilly, Pauline Meskell, Catriona Kennedy, Alice Coffey, Isabelle Delaunois, Liz Dore, Siobhan Howard, Sheila Ryan.

**Formal analysis:** Pauline O'Reilly, Pauline Meskell, Barbara Whelan, Catriona Kennedy, Alice Coffey, Donal G. Fortune, Donna M. Wilson, Siobhan Howard.

**Funding acquisition:** Pauline O'Reilly, Bart Ramsay, Sheila Ryan.

**Investigation:** Bart Ramsay.

**Methodology:** Pauline O'Reilly, Pauline Meskell, Barbara Whelan, Catriona Kennedy, Alice Coffey, Donal G. Fortune, Donna M. Wilson, Siobhan Howard, Sheila Ryan.

**Project administration:** Pauline O'Reilly.

**Supervision:** Pauline O'Reilly.

**Validation:** Pauline O'Reilly.

**Writing – original draft:** Pauline O'Reilly.

**Writing – review & editing:** Pauline O'Reilly, Pauline Meskell, Barbara Whelan, Catriona Kennedy, Bart Ramsay, Alice Coffey, Donal G. Fortune, Sarah Walsh, Saskia Ingen-Housz-Oro, Christopher B. Bunker, Donna M. Wilson, Isabelle Delaunois, Liz Dore, Siobhan Howard, Sheila Ryan.

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
