## [Decision Letter · Decision Letter 0]

11 Mar 2022

PONE-D-21-37340Psychotherapeutic interventions for burns patients and the potential use with Stevens-Johnson syndrome and toxic epidermal necrolysis patients: a systematic integrative reviewPLOS ONE

Dear Dr. O'Reilly,

Thank you for submitting your manuscript to PLOS ONE. After careful consideration, we feel that it has merit but does not fully meet PLOS ONE’s publication criteria as it currently stands. Therefore, we invite you to submit a revised version of the manuscript that addresses the points raised during the review process.

Editor - Thank you for submitting your paper to us for review.  I sent it to four distinguished referees for comment and decision of whom two agreed to review; you will see these below.  They thought that the paper has merit, but each have raised some substantial issues to be addressed in a revision.  Please carefully consider the comments below and reply directly to each in a cover letter with appropriate marked and linked changes to the manuscript.  I look forward to seeing the revision, which I will send back to the same referees for further comment and decision.  Please understand that this is not a guarantee of future publication, as the revision must stand on its own merit.

We look forward to receiving your revised manuscript.

Kind regards,

Steven Eric Wolf, MD

Academic Editor

PLOS ONE

Journal Requirements:

Reviewers' comments:

Reviewer's Responses to Questions

**Comments to the Author**

1. Is the manuscript technically sound, and do the data support the conclusions?

Reviewer #1: Yes

Reviewer #2: Yes

2. Has the statistical analysis been performed appropriately and rigorously? 

Reviewer #1: N/A

Reviewer #2: Yes

3. Have the authors made all data underlying the findings in their manuscript fully available?

Reviewer #1: Yes

Reviewer #2: Yes

4. Is the manuscript presented in an intelligible fashion and written in standard English?

Reviewer #1: Yes

Reviewer #2: Yes

5. Review Comments to the Author

Reviewer #1: Thankyou for allowing me to review the manuscript entitled: Psychotherapeutic interventions for burns patients and the potential use with Stevens-Johnson syndrome and toxic epidermal necrolysis patients: a systematic integrative review. This is an interesting review and how it was conducted. The question that comes to mind is does it add anything new to the body of knowledge as SJS/TEN patients are often cared for a burn intensive care unit as SJS/TEN mirrors the requirements of burn patients due to insensible losses through wounds, however fluid requirements are reduced in burn patients with similar TBSA involvement.

The authors use both PTS and PTSD. For consistency use one acronym.

The referencing formatting intext needs review as shown here: [39, 45, 47] [40, 41, 46, 48, 50] [42, 49].

There is a missing arrow on the SJS/TEN PRISMA.

Tables are difficult to read due to portrait presentation. Tables need to be formatted accordingly in Landscape. The author, year and country should be combined in one column. The first column with the numbers should be deleted.

Strongly suggest the authors review how other review tables are presented.

Some key literature has been missed and should be included for the currency of the review:

Schneider, J. A., & Cohen, P. R. (2017). Stevens-Johnson Syndrome and Toxic Epidermal Necrolysis: A Concise Review with a Comprehensive Summary of Therapeutic Interventions Emphasizing Supportive Measures. Advances in therapy, 34(6), 1235–1244. https://doi.org/10.1007/s12325-017-0530-y

Shanbhag, S. S., Chodosh, J., Fathy, C., Goverman, J., Mitchell, C., & Saeed, H. N. (2020). Multidisciplinary care in Stevens-Johnson syndrome. Therapeutic Advances in Chronic Disease. https://doi.org/10.1177/2040622319894469

Frantz, R., Huang, S., Are, A., & Motaparthi, K. (2021). Stevens-Johnson Syndrome and Toxic Epidermal Necrolysis: A Review of Diagnosis and Management. Medicina (Kaunas, Lithuania), 57(9), 895. https://doi.org/10.3390/medicina57090895

Reviewer #2: Review – Psychotherapeutic interventions for burn patients and the potential use with SJS/TENs patients: a systematic integrative review.

To the best of my understanding the criteria have been met:

1. The study presents the results of original research.

2. Results reported have not been published elsewhere.

3. Experiments, statistics, and other analyses are performed to a high technical standard and are described in sufficient detail.

4. Conclusions are presented in an appropriate fashion and are supported by the data.

5. The article is presented in an intelligible fashion and is written in standard English.

6. The research meets all applicable standards for the ethics of experimentation and research integrity.

7. The article adheres to appropriate reporting guidelines and community standards for data availability.

This systematic integrative review assesses the published evidence of psychotherapeutic interventions to alleviate psychological distress following burn injury, in the absence of published evidence of psychotherapeutic interventions for Stevens-Johnson syndrome and TEN, on the basis that is it reasonable to suggest that the psychological sequelae of the condition are similar.

This is a well written and clearly presented manuscript, which I believe meets the journal criteria and can be accepted for publication with some minor changes. I like the fact that you were able to include papers from the French language as well as English.

Abstract Aims and objectives

Add in the word ‘adult’

The aim of this systematic integrative review was to synthesize the evidence relating to psychotherapeutic interventions used with adult burns patients and patients with SJS/TENs.

Abstract Results

Remove the word ‘dearth’ and clearly state that there was no evidence for psych interventions in SJS/TENs.

Introduction

Mostly throughout the manuscript the intext reference falls after the full stop at the end of a sentence or after a comma mid-sentence instead of before the punctuation; eg line 90, 98, 105, 115. Please correct.

Methods

The review question should state burn patient ‘either/or’ SJS-TENs patients not ‘both/and’:

‘What is the evidence on the psychotherapeutic 150 interventions which have been used with (either) both adult burns patients or and patients with 151 SJS/TEN, during the acute stage of the illness, to reduce symptoms aligned to PTS and 152 improve quality of life?’.

State how you completed the QA process as a group and list author initials.

Table 2: A lot of the information can be reduced without losing context. I would probably present the qualitative studies in separate table, purely because the information presented is wordier and therefore from a formatting point of view will look better separate. Perhaps reduce the font/line separation etc similar to table 3 to allow the columns to be read more easily, although this might be an editing task by the journal.

• Do you need paper number in column 1?

• Add the country under the authors names and add the reference

• Summarise the aim

• Add in outcome measures used - this column appears to be blank for all papers. Would bbe interesting to add the time post-burn at which the outcomes are measured.

• Summarise key findings (with stats if available).

• Reorder columns: Author/ country; aim; study design; study participants; outcome measures; analyses; key findings.

Line 253: Please change: ‘Burn Specific Health Scale – Brief’ not ‘Burning Specific Health Scale’

Discussion:

You are interested in long term psychological sequelae after SJS/TENs but not many of these studies assesses the long term effects of the interventions found, with some exceptions eg patients who receive the multimedia training are assessed at 3 and 6 months post intervention. In fact, theme 2 treatments seem to address the long term recovery more than the theme 1 treatments. This would be good to explore in the discussion, particularly with regards to the need for longitudinal research in the future.

I have no competing interests to declare.

Well done, this is looking good. Hopefully my suggestions will take it up a notch, too.

6. PLOS authors have the option to publish the peer review history of their article (what does this mean?). If published, this will include your full peer review and any attached files.

Reviewer #1: No

Reviewer #2: No

---

## [Author Response · Author response to Decision Letter 0]

21 Apr 2022

Reviewers 

PLOSONE

April 21st, 2022

Re: Paper Re-Submission

PONE-D-21-37340

Psychotherapeutic interventions for burns patients and the potential use with Stevens-Johnson syndrome and toxic epidermal necrolysis patients: a systematic integrative review

Dear Reviewers,

I wish to thank you both for taking the time to review our systematic review. 

We found the feedback and comments to be most helpful and insightful. In light of these, we have revised the manuscript. Please see below, a supporting document outlining the details of the revisions. 

Once again, many thanks for your comments and advice. I look forward to hearing from you.

Yours sincerely

Dr Pauline O’Reilly

Senior Lecturer

Department of Nursing and Midwifery,

University of Limerick,

Limerick,

Ireland 

Email: Pauline.OReilly@ul.ie

On behalf of the research team

Table of Revision

Reference number: PONE-D-21-37340 Psychotherapeutic interventions for burns patients and the potential use with Stevens-Johnson syndrome and toxic epidermal necrolysis patients: a systematic integrative review. PLOS ONE

Editor Response

Changes to Data Availability statement

We note that you have stated that you will provide repository information for your data at acceptance. Should your manuscript be accepted for publication, we will hold it until you provide the relevant accession numbers or DOIs necessary to access your data. If you wish to make changes to your Data Availability statement, please describe these changes in your cover letter and we will update your Data Availability statement to reflect the information you provide. The article was checked to ensure it met PLOSONE requirements. Headings were revised and figures were uploaded unto PACE and now meet the journal requirements. All references were reviewed and amended where necessary, to meet PLOSONE referencing criteria. 

In relation to the Data Availability statement, please amend to -all material relating to this systematic review are available within the article and supporting information files. 

Reviewer: #1 Response

Thank you for allowing me to review the manuscript entitled: Psychotherapeutic interventions for burns patients and the potential use with Stevens-Johnson syndrome and toxic epidermal necrolysis patients: a systematic integrative review. This is an interesting review and how it was conducted. The question that comes to mind is does it add anything new to the body of knowledge as SJS/TEN patients are often cared for a burn intensive care unit as SJS/TEN mirrors the requirements of burn patients due to insensible losses through wounds, however fluid requirements are reduced in burn patients with similar TBSA involvement.

 Included referenced evidence on this point, see Page 5&6, Line 192-209

The authors use both PTS and PTSD. For consistency use one acronym. PTS was used when a diagnostic instrument wasn’t used (the patients had symptoms of post-traumatic stress, but they were not sufficient to meet the full criteria for a disorder of PTSD). 

PTSD was used when a diagnosis had been given using a recognised instrument (the symptoms are of sufficient intensity and frequency to warrant a diagnosis - meeting the accepted criteria, whether that is according to DSM or ICD criteria, DSM or ICD - depending in the country/continent where the research was conducted).

The referencing formatting intext needs review as shown here: [39, 45, 47] [40, 41, 46, 48, 50] [42, 49]. All references were reviewed and amended where necessary, to meet PLOSONE referencing criteria.

There is a missing arrow on the SJS/TEN PRISMA. Amended – arrow included

Tables are difficult to read due to portrait presentation. Tables need to be formatted accordingly in Landscape. The author, year and country should be combined in one column. The first column with the numbers should be deleted. 

Strongly suggest the authors review how other review tables are resented. The tables were revised and are presented in Landscape format. In relation to Tables 2 & 3 - the first column was deleted, and the author, year and country were combined into one column. Please see amended tables, Pages 12-17

Some key literature has been missed and should be included for the currency of the review:

• Schneider, J. A., & Cohen, P. R. (2017). Stevens-Johnson Syndrome and Toxic Epidermal Necrolysis: A Concise Review with a Comprehensive Summary of Therapeutic Interventions Emphasizing Supportive Measures. Advances in therapy, 34(6), 1235–1244. https://doi.org/10.1007/s12325-017-0530-y

• Shanbhag, S. S., Chodosh, J., Fathy, C., Goverman, J., Mitchell, C., & Saeed, H. N. (2020). Multidisciplinary care in Stevens-Johnson syndrome. Therapeutic Advances in Chronic Disease. https://doi.org/10.1177/2040622319894469

• Frantz, R., Huang, S., Are, A., & Motaparthi, K. (2021). Stevens-Johnson Syndrome and Toxic Epidermal Necrolysis: A Review of Diagnosis and Management. Medicina (Kaunas, Lithuania), 57(9), 895. https://doi.org/10.3390/medicina57090895 Amended. This key literature has been included in the Introduction and the Discussion. Please see:

Page 5; Line 173-175.

Page 5&6; Line 192-209.

Page 30&31; Line 943-963

Reviewer: #2 

Abstract Aims and objectives:

Add in the word ‘adult’

The aim of this systematic integrative review was to synthesize the evidence relating to psychotherapeutic interventions used with adult burns patients and patients with SJS/TENs. Amended. The word ‘adult’ was included in the aim. Please see: Page 2, Line 49-51

Abstract Results

Remove the word ‘dearth’ and clearly state that there was no evidence for psych interventions in SJS/TENs. Amended. The word ‘Dearth’ was replaced with ‘no evidence’. Please see:

Page 3, Line 72

Mostly throughout the manuscript the intext reference falls after the full stop at the end of a sentence or after a comma mid-sentence instead of before the punctuation; eg line 90, 98, 105, 115. Please correct. All references were reviewed and amended where necessary, to meet PLOSONE referencing criteria. All intext references are placed before punctuation. 

Methods: The review question should state burn patient ‘either/or’ SJS-TENs patients not ‘both/and’:

‘What is the evidence on the psychotherapeutic interventions which have been used with (either) both adult burns patients or and patients with SJS/TEN, during the acute stage of the illness, to reduce symptoms aligned to PTS and improve quality of life?’ This review question was revised. 

Please see:

Page 8, Line 248

State how you completed the QA process as a group and list author initials. This section was amended to include the process of assessment and the initials of the authors who appraised the studies. 

Please see:

Page 10, Line 300-305.

Table 2: A lot of the information can be reduced without losing context. I would probably present the qualitative studies in separate table, purely because the information presented is wordier and therefore from a formatting point of view will look better separate. Perhaps reduce the font/line separation etc similar to table 3 to allow the columns to be read more easily, although this might be an editing task by the journal.

• Do you need paper number in column 1?

• Add the country under the authors names and add the reference

• Summarise the aim

• Add in outcome measures used - this column appears to be blank for all papers. Would be interesting to add the time post-burn at which the outcomes are measured.

• Summarise key findings (with stats if available).

• Reorder columns: Author/ country; aim; study design; study participants; outcome measures; analyses; key findings. The table was amended as per suggestions.

There are now two tables viz. Table 2 presents the quantitative papers and Table 3 presents the qualitative papers. 

Column 1 was removed.

The country, authors’ names and reference in one column.

All aims were summarized.

Outcome measures and references were added. however, it was difficult to determine the time post burn at which the outcomes were measured for most of the studies.

Key findings were summarized, and statistics were included were available.

Columns were reordered as per recommendation.

Please see Table 2&3:

Pages 12-17

Line 253: Please change: ‘Burn Specific Health Scale – Brief’ not ‘Burning Specific Health Scale’ Spelling error amended from Burning to Burn. Please see:

Page 19, Line 515. 

8.Discussion:

You are interested in long term psychological sequelae after SJS/TENs but not many of these studies assesses the long-term effects of the interventions found, with some exceptions e.g., patients who receive the multimedia training are assessed at 3- and 6-months post intervention. In fact, theme 2 treatments seem to address the long-term recovery more than the theme 1 treatments. This would be good to explore in the discussion, particularly with regards to the need for longitudinal research in the future. A section was included outlining the need for more longitudinal studies. Please see:

Page 30&31, Line 962-982

---

## [Decision Letter · Decision Letter 1]

16 May 2022

PONE-D-21-37340R1Psychotherapeutic interventions for burns patients and the potential use with Stevens-Johnson syndrome and toxic epidermal necrolysis patients: a systematic integrative reviewPLOS ONE

Dear Dr. O'Reilly,

Thank you for submitting your manuscript to PLOS ONE. After careful consideration, we feel that it has merit but does not fully meet PLOS ONE’s publication criteria as it currently stands. Therefore, we invite you to submit a revised version of the manuscript that addresses the points raised during the review process.

We look forward to receiving your revised manuscript.

Kind regards,

Steven Eric Wolf, MD

Academic Editor

PLOS ONE

Journal Requirements:

Additional Editor Comments:

Editor - Thank you for resubmitting your paper. As promised, I sent it back to the original referees who are now almost completely satisfied save a few minor issues. Please carefully consider the comments below and reply directly to each in a cover letter with appropriate marked and linked changes to the manuscript. I look forward to receiving the next version which I will handle personally for timeliness.

Reviewers' comments:

Reviewer's Responses to Questions

**Comments to the Author**

1. If the authors have adequately addressed your comments raised in a previous round of review and you feel that this manuscript is now acceptable for publication, you may indicate that here to bypass the “Comments to the Author” section, enter your conflict of interest statement in the “Confidential to Editor” section, and submit your "Accept" recommendation.

Reviewer #1: All comments have been addressed

Reviewer #2: All comments have been addressed

2. Is the manuscript technically sound, and do the data support the conclusions?

Reviewer #1: Yes

Reviewer #2: Yes

3. Has the statistical analysis been performed appropriately and rigorously? 

Reviewer #1: I Don't Know

Reviewer #2: Yes

4. Have the authors made all data underlying the findings in their manuscript fully available?

Reviewer #1: Yes

Reviewer #2: Yes

5. Is the manuscript presented in an intelligible fashion and written in standard English?

Reviewer #1: Yes

Reviewer #2: Yes

6. Review Comments to the Author

Reviewer #1: Thank you for allowing me to re-review the manuscript entitled: Psychotherapeutic interventions for burns patients and the potential use with Stevens Johnson syndrome and toxic epidermal necrolysis patients: A systematic integrative review. Thank you for addressing the comments raised in the previous review.

PTSD and PTS are not presented or defined as per formatting requirements as all acronyms need to be formatted accordingly – Post Traumatic Stress Disorder (PTSD) when used for the first time then PTSD thereafter.

Line 134 there are 2 full stops in at the end of the sentence.

There is inconsistency with how et al is presented when the author is used first et al., & et al. is used throughout.

viz is throughout the manuscript no idea what it is referring to as could not find an explanation.

When using direct quotes from the included qualitative studies, I believe a page number may be required?

Reviewer #2: (No Response)

7. PLOS authors have the option to publish the peer review history of their article (what does this mean?). If published, this will include your full peer review and any attached files.

Reviewer #1: No

Reviewer #2: No

---

## [Author Response · Author response to Decision Letter 1]

19 May 2022

Table of Revision

Reference number: PONE-D-21-37340R1 Psychotherapeutic interventions for burns patients and the potential use with Stevens-Johnson syndrome and toxic epidermal necrolysis patients: a systematic integrative review. PLOS ONE

Journal Requirements

Please review your reference list to ensure that it is complete and correct. If you have cited papers that have been retracted, please include the rationale for doing so in the manuscript text or remove these references and replace them with relevant current references. Any changes to the reference list should be mentioned in the rebuttal letter that accompanies your revised manuscript. If you need to cite a retracted article, indicate the article’s retracted status in the References list and also include a citation and full reference for the retraction notice.

Response

The article was checked to ensure it met PLOSONE requirements. In the previous revised article, all references were reviewed and amended where necessary, to meet PLOSONE referencing criteria. No reference was retracted. With the tracked changes, some references are under the heading ‘deleted’, however, the corrected versions of the references were updated and included. All references were rechecked. 

Reviewer: #1

PTSD and PTS are not presented or defined as per formatting requirements as all acronyms need to be formatted accordingly – Post Traumatic Stress Disorder (PTSD) when used for the first time then PTSD thereafter.

Response

Amended: Post-Traumatic Stress Disorder was written in full initially and the acronym PTSD was used subsequently. 

Please see Pg 5; Line 111

Post Traumatic Symptoms had already been written in full and the acronym PTS used thereafter.

Please see Pg 4; Line 97-98

Reviewer: #1

Line 134 there are 2 full stops in at the end of the sentence.

Response

Amended: Extra full stop removed.

Please see Pg 5; Line 125 

Reviewer: #1

There is inconsistency with how et al is presented when the author is used first et al., & et al. is used throughout.

Response

Amended: The inconsistency with et al., was revised and amended. There were 18 revisions. Please see Pg 4; Line 94; Pg 4; Line 101; Pg 19; Line 265; Pg 19; Line 271; Pg 20; Line 279; Pg 21; Line 307; Pg 22; Line 341 – 348; Pg 24; Line 373 – 376; Pg 25; Line 406 & 410; Pg 28; Line 499 and Pg 32; Line 577.

Reviewer: #1

viz is throughout the manuscript no idea what it is referring to as could not find an explanation. 

Response

Amended – the use of ‘vis.’ was revised throughout the article and removed were appropriate.

Please see Pg 18; Line 246; Pg 20; Line 279; Pg 26; Line 433; Pg 28; Lines 470, 483, 487, Pg 29; Line 498 & 508; Pg 30; Line 530 and Pg 31; Line 553. 

Reviewer: #1

When using direct quotes from the included qualitative studies, I believe a page number may be required? 

Response

Amended: The page number was used with each quotation.

Please see Pg 27; Line 457; Pg 27; Line 461 and Pg 27; Line 468.

Thank you.

---

## [Decision Letter · Decision Letter 2]

10 Jun 2022

Psychotherapeutic interventions for burns patients and the potential use with Stevens-Johnson syndrome and toxic epidermal necrolysis patients: a systematic integrative review

PONE-D-21-37340R2

Dear Dr. O'Reilly,

We’re pleased to inform you that your manuscript has been judged scientifically suitable for publication and will be formally accepted for publication once it meets all outstanding technical requirements.

Kind regards,

Steven Eric Wolf, MD

Academic Editor

PLOS ONE

Additional Editor Comments (optional):

Reviewers' comments:

Reviewer's Responses to Questions

**Comments to the Author**

1. If the authors have adequately addressed your comments raised in a previous round of review and you feel that this manuscript is now acceptable for publication, you may indicate that here to bypass the “Comments to the Author” section, enter your conflict of interest statement in the “Confidential to Editor” section, and submit your "Accept" recommendation.

Reviewer #1: All comments have been addressed

Reviewer #2: All comments have been addressed

2. Is the manuscript technically sound, and do the data support the conclusions?

Reviewer #1: Yes

Reviewer #2: Yes

3. Has the statistical analysis been performed appropriately and rigorously? 

Reviewer #1: I Don't Know

Reviewer #2: Yes

4. Have the authors made all data underlying the findings in their manuscript fully available?

Reviewer #1: Yes

Reviewer #2: Yes

5. Is the manuscript presented in an intelligible fashion and written in standard English?

Reviewer #1: Yes

Reviewer #2: Yes

6. Review Comments to the Author

Reviewer #1: Thank you for allowing me to re review this manuscript entitled Psychotherapeutic interventions for burns patients and the potential use with Stevens-Johnson syndrome and toxic epidermal necrolysis patients: a systematic integrative review. No further comments.

Reviewer #2: (No Response)

7. PLOS authors have the option to publish the peer review history of their article (what does this mean?). If published, this will include your full peer review and any attached files.

Reviewer #1: No

Reviewer #2: No

---

## [Editor Report · Acceptance letter]

16 Jun 2022

PONE-D-21-37340R2 

Psychotherapeutic interventions for burns patients and the potential use with Stevens-Johnson syndrome and toxic epidermal necrolysis patients: a systematic integrative review 

Dear Dr. O'Reilly:

I'm pleased to inform you that your manuscript has been deemed suitable for publication in PLOS ONE. Congratulations! Your manuscript is now with our production department. 

Kind regards, 

on behalf of

Dr. Steven Eric Wolf 

Academic Editor

PLOS ONE